# Assessing the ecological validity of soundscape reproduction in different laboratory settings

**Cynthia Tarlao** [1,2] *, **Daniel Steele** [1,2], **Catherine Guastavino** [1,2]

**1** Multimodal Interaction Laboratory, School of Information Studies, McGill University, Montreal, Quebec, Canada, **2** Centre for Interdisciplinary Research in Music Media and Technology, Montreal, Quebec, Canada

* cynthia.tarlao@mail.mcgill.ca

## Abstract

The ever-growing body of soundscape research includes studies conducted both in everyday life environments and in laboratory settings. Yet, laboratory settings differ from in-situ and therefore may elicit different perceptions. The present study explores the *ecological validity* of soundscape reproduction in the laboratory using first-order Ambisonics and of different modes of questionnaire administration. Furthermore, it investigates the influence of the contextual factors of time of day, day of the week, and location on site on soundscape evaluations *in situ* and in the laboratory, based on the Swedish Soundscape Quality Protocol. We first tested measurement invariance between the computer-based and pen-and-paper administration of the soundscape questionnaire. We then investigated the influence of the above-mentioned contextual factors on soundscape evaluations, as well as the effect of stimuli selection in the laboratory. The analyses confirmed the underlying dimensions of proposed soundscape assessment questionnaires, confirmed metric invariance between computer and pen-and-paper, and revealed significant influences of time, day, and location on soundscape scales. This research represents a critical step in rigorously assessing soundscape evaluations in the laboratory and establishes solid evidence for the use of both *in situ* and laboratory soundscape studies.

## Introduction

### State-of-the-art in soundscape research

A growing body of literature on urban soundscape has emerged to contrast urban noise mitigation [1]. Soundscape, defined as the "acoustic environment as perceived or experienced and/ or understood by a person or people, in context" [2], affords new strategies for urban sound management by focusing on human experience and considering sound as a resource rather than a nuisance. In this way, the soundscape approach offers opportunities for cities, both to improve urban experiences from the earliest stages of urban design, and to work on the development of new overarching policies [1].

**Data Availability Statement:** Due to the nature of consent obtained from participants, data cannot be made available outside of the research team, as per the ethics approval obtained from the McGill Research Ethics Board. For future data requests,

please contact the McGill Ethics Officer at lynda.
mcneil@mcgill.ca.

**Funding:** The studies in this paper were funded by grants from Canada's Social Sciences and Humanities Research Council (https://www.sshrc-crsh.gc.ca/) [#430-2016-01198 and #890-2017-0065 to CG, Sounds in the City]. The writing of this paper was supported by research grant NSERC RGPAS-2019-00035 from the Natural Sciences and Engineering Research Council of Canada (https://www.nserc-crsng.gc.ca/) to CG. The funders had no role in study design, data collection and analysis, decision to publish, or preparation of the manuscript.

**Competing interests:** The authors have declared that no competing interests exist.

As the ISO definition [2] suggests, soundscape research takes context into account as a critical factor in how humans perceive acoustic environments. As such, models and frameworks of the contextually-mediated relationships between soundscape and listener have started to emerge in the field [1, 3, 4], with increased interest in the last few years [5]. Specifically of interest to us here are the spatiotemporal factors mentioned in the ISO: "The context includes the interrelationships between person and activity and place, *in space and time*" [2], based on their theorization as important influences in activity-centric soundscape frameworks [1, 3, 6].

Urban soundscapes have been shown to vary in loudness and dominant sources as a function of time of day [7, 8], day of the week (weekday vs. weekend), and location within the studied space [9]. However, most spatiotemporal factors examined in the literature revolve around city- or neighborhood-level location (i.e., functions of spaces) [7, 8], spatial characteristics of the studied space and user behavior (frequency, duration, preference) [10], or short-term temporality [11]. In this paper, we will be focusing on the spatial factor of location within the studied space, and the temporal factors of day of the week and time of day.

**Soundscape assessment instruments.** In the last decade, several soundscape measurement scales have been developed and refined to elicit human evaluations of acoustic environments (see [12] for a methodological review). The Swedish Soundscape Quality Protocol (SSQP), developed in Swedish and English in lab-based experiments [13], measures soundscape evaluations along the two main dimensions of *pleasantness* and *eventfulness* and formed an important basis for the ISO standard on soundscape methodologies [14]. In response to urban studies considerations [15], Axelsson [16] proposed adding the dimension of soundscape *appropriateness* to the SSQP, which is understood as soundscape *appropriateness* for specific activities [17] in this study. In addition, the soundscape approach considers sound as a resource, most notably in terms of the potential for restoration [18] provided by urban soundscapes [19], with the recent development of the Perceived Restorativeness Soundscape Scale [20].

In parallel to the growing interest in the soundscape approach to improve urban experiences and policies [1], researchers are exploring the potential of soundscape simulation and manipulation in the laboratory. Laboratory experiments reproducing soundscapes in virtual acoustics allow for more control than *in situ* studies, especially regarding the ability to manipulate variables and explore causal relationships between them. Such increased control can come with other costs, such as artefacts introduced by the recording and reproduction techniques, an altered sensory and cognitive experience, or a lack of certain contextual factors like time of day or weather, which all contribute to limiting the transferability of findings to other contexts. Nevertheless, virtual soundscapes offer flexibility in posing new research questions for the academic community and could, in the longer term, provide opportunities for the urban design and planning communities to "visualize", understand, manipulate, or communicate soundscapes.

## Soundscape reproduction

Soundscape researchers are conscious of the gap between academic progress and urban practice [21]. One important aspect of this gap is a lack of tools that are easy to use and useful for integrating sound considerations in the practice of urban professionals. Soundscape researchers therefore understand that a major avenue to bridge the research-practice gap lies in the development of soundscape simulation tools [21]. Such simulation tools would offer technological support for urban professionals to understand and imagine ("sketch out") soundscapes. However, urban professionals have not benefited from this research up until now, because of both the lack of accessible content for non-experts [22] and the complexity of use of most

audio manipulation technologies [1]. Nevertheless, urban professionals have shown interest in 3D-reproduction of soundscapes for knowledge mobilization, for the purpose of learning from sound experts and researchers, for its immersive benefits, including "emphasiz[ing] the importance of human experience" and "show[ing] tangible design potential" [23]. In the same spirit, it is no leap of the imagination to consider the potential applications to support the integration of soundscape in the design process, from design to communication of designs to stakeholders [1].

There already exists a number of commercial tools for acoustically accurate 3D simulation of soundscapes by environmental acoustics experts hired by urban professionals, such as MithraSound [24]. Such tools make use of the technical specifications of existing and envisaged urban designs to produce accurate modeling of sound sources and sound propagation but require a high degree of expertise to wield. In contrast, different soundscape simulator tools have been developed by researchers, most often for experimental testing both with research purposes and with applied goals of defining city users' preferences in the context of specific development projects. However, most of those technologies have not been developed with the urban professional in mind, and this lack of tools to help urban professionals understand the quality of soundscapes limits their ability to consider sound beyond the required acoustic measures [1], even when they readily understand sound can be a resource for their own practice [22].

The premise at the root of this study stems from the idea that research-oriented spatial soundscape simulation tools, which generally aim less for physical accuracy than to center perception and experience, could offer a foundation for the development of practical applications for urban professionals with moderate changes to account for their needs. Note that these tools are not intended to displace acoustics expertise, but to complement it. In the visual analog, designers will often provide models and collages that lead to more precise CAD drawings done in collaboration with engineers.

**Existing tools for soundscape reproduction.**    Two main uses can be distinguished for soundscape reproduction applications: first, as-is reproduction and databases, which rely directly on existing recording and reproduction techniques; and second, what can be called *simulators*, which include some level of interactive manipulation of the reproduced soundscape and sound sources. Soundscape composition has also been used for more artistic purposes (e.g., [25]), but we will not explore this aspect here. This paper makes use of the former and what follows will be a quick non-exhaustive overview of some of those direct reproduction tools in research.

Soundscape researchers have shown increasing interest in using spatial sound reproduction to study soundscape ratings in the laboratory in the last decade, starting with Brambilla and Maffei [26], who created visual and audio design scenarios for two Neapolitan public squares to explore the potential of laboratory simulation of design changes. Interestingly, they found that the sound component always had more influence than the visuals on the overall assessment. More recently, in the same vein of designing scenarios, a French team [27] composed immersive sound scenes, from recordings of outdoor spaces' backgrounds and isolated vehicles, to study noise annoyance, for which they obtained high realism ratings, although no "real life" comparisons could understandably be conducted.

For research using soundscape reproduction as is, a recent example is a study comparing soundscape ratings *in situ* and in the laboratory in order to establish a model of the factors influencing soundscape ratings [28], which showed no differences of the overall pleasantness between the *in situ* soundwalk and the laboratory immersive reproduction. This team also found a higher correlation of overall pleasantness with soundscape pleasantness than visual pleasantness. In the same vein, a Croatian study was conducted to test the influence of sound

art installations in public spaces with a "virtual soundwalk" in the laboratory [29]. This "virtual soundwalk" methodology [30] uses 3D sound recordings and 2D panoramic pictures at fixed locations, defined by the researchers, reproduced sequentially in the laboratory, for participants to evaluate. Through comparisons of participant mean ratings of the SSQP between *in situ* and laboratory settings, the authors concluded that the "virtual soundwalk" yielded ratings similar to those collected *in situ*, thus validating the methodology. However, the authors did not provide any statistical analysis to substantiate this claim.

With both research goals and urban design applications in mind, the Urban Soundscapes of the World project compiled a comprehensive database of audio-visual recordings of systematically-selected urban sites from cities all over the world [31] to offer a wide range of urban soundscapes for perceptual experiments. In the same spirit of supporting research and creative practice, CityTones is a repository of soundscapes captured using 360º audio and video or photo, with both recording and labelling partially crowdsourced [32]. For an application with a more popular goal, *I Hear NY3D*, a project for capturing and reproducing 3D soundscapes in New York City, collected 3D recordings of various locations in Manhattan [33] to offer an interactive interface to experience the soundscapes of Manhattan virtually. Those awareness-raising efforts are essential to archive urban soundscapes for use in research and creative practice. The question remains open, however, as to how the use of such systems could facilitate the work of professionals of the built environment, such as urban designers, architects, etc.

In this work, the chosen method of reproduction will be Ambisonics. The Ambisonic technique [34, 35] is most commonly used in research and now also being implemented in widespread commercial-consumer applications such as Youtube 360˚. This technique can be presented on any playback configuration and prioritizes envelopment and immersion over precise localization of sound sources [36], elicit similar cognitive processes to *in situ* results especially in relation to urban background noise [37].

When conducting laboratory experiments, one should be aware of their limitations in terms of the perceptual and cognitive processes being studied. Laboratory settings differ from everyday life situations and therefore may elicit different judgments, whether through different perceptions, experiences, expectations, or biases, specifically in terms of contextual factors (for example, the reason for choosing to visit a particular space at a particular time). This is an important tenet of experimental psychology, known as *ecological validity*. The ecological validity of data collected with spatial audio in laboratory settings has become a common matter of interest and concern for soundscape researchers [21].

## Ecological approach

The concept of ecological validity was first introduced by Egon Brunswik [38, 39] and later developed into the concept we understand today by James Gibson [40], both psychologists investigating visual perception. As Brunswik [38] first stated, perception of our environment is ambiguous, with multiple "probable partial causes", and requires compromises between informative environmental cues to determine a "best bet" on the perception of an object. This "intrinsic lack of perfection" in everyday life should not be eschewed by the experimenter and the experiment should be designed to present "conditions representative of actual life".

Gibson is possibly better known than Brunswik for having developed the *ecological approach to visual perception* [40] which is now the more common understanding of ecological validity and has been accepted by psychology textbooks: "Studies are high in ecological validity if the conditions in which the research is conducted are similar to the natural setting where the results will be applied" [41].

The ecological validity of an experimental design rests on three elements: 1) the participants being representative of the population the results are intended to be generalized to; 2) the experimental conditions being representative of the actual conditions the results are meant to apply to; and 3) the task (including instructions and data collection instruments) eliciting similar cognitive processes than in the everyday life situations [38, 42]. Only then can the experimenter ensure that the research design is ecologically valid, that is, that it truly allows to explore the cognitive processes of the everyday-life conditions it purports studying. This also means that it is often not possible to know in advance the extent to which a research design will be ecologically valid or not [43]. Even with sound theory based on previously accepted arguments and experiments, new designs need to be validated for the population, conditions, and cognitive processes they intend to represent.

**Ecological approach to soundscape.** The ecological approach was first applied to auditory perception with VanDerveer's [44] work exploring the perception of environmental sounds. Gaver's work is also significant for defining the notion of *everyday listening*, in contrast to musical listening [45, 46] however much of it was embedded in the perception of *physical dimensions*, in relation to materials (i.e., liquids, solids, gasses) and simple events (e.g., impact, scraping, gust). This approach did not take into account higher cognitive processes of socially-constructed meaning and memory [47] which play a critical role for complex everyday sounds. Indeed, Dubois [48] showed that sounds are also perceived and identified holistically by listeners who integrate everyday situations in which the sounds are experienced into complex mental representations [49]. Dubois further discusses the methodological consequences for investigating everyday cognition in laboratory settings, including reconsidering the opposition of subjective and objective [50], recalling the arguments put forth by Brunswik [38, 39] and Gibson [40].

More recently, Guastavino [42] reviewed studies that explored the three aspects of ecological validity of auditory perception of reproduced urban soundscapes. Regarding participants, sound experts (sound engineers) and non-experts (city users) focused on different aspects of the soundscape reproduction, highlighting the relationship between individual experience and ecological validity [51]. Non-experts attended to the scene holistically, preferring the feeling of immersion over precision of the reproduced scene, whereas experts prioritize precision and stability in a more analytical listening strategy. Regarding condition representativeness, they found that different reproduction methods and systems were preferred depending on the soundscape reproduced [37, 52]. For example, speaker configurations including a subwoofer were found more realistic only for recordings of traffic noise. Another example is that soundscapes where sounds were expected to come from above were judged as more realistic when reproduced over a 3D configuration, while 1D and 2D configurations were found more realistic for soundscapes where sounds needed to be clear and localizable. Examples of 1D, 2D, and 3D loudspeaker configurations include, respectively, a stereo setup for sounds positioned in the left-right dimension, a ring of loudspeakers around the listener with sounds spatialized on the horizontal plane, and a sphere of loudspeakers presenting sound spatialized horizontally and vertically. These results highlight the importance of choosing a reproduction system valid for the specific sounds and soundscapes (conditions) studied [52], as well as for who is evaluating them. The principle here is to make sure the information reproduced generates perceptual judgments as close to the everyday-life soundscape would [42]. A more recent study combining spatial audio and video recordings found no significant differences between *in situ* and 2D Ambisonic reproduction in terms of SSQP ratings and dominant sound sources [53].

Finally, in terms of the experimental process, Guastavino showed that different reproduction systems prompted different cognitive representations [37]. In the case of soundscape reproduction, source identification and spatial immersion, especially as it contributes to the cognitive representation of city background noise, might be most important. 3D multichannel

configurations were found to offer the best spatial immersion, while source identification remained close to everyday life situations. Hong et al. [53] also found a 2D Ambisonic reproduction method to elicit significantly higher immersion, realism, externalization, and listening experience ratings than Ambisonics-based binaural reproduction methods.

Another aspect of the experimental process is the procedure. Among several decisions, the experimenter must choose how the data collection instruments (e.g., questionnaire) will be administered. For instance, a NASA study [54] found that the mode of administration of their task load index scales (NASA-TLX) influenced the results. On average, results obtained on computer were significantly higher than those obtained with a paper-and-pen method, although the patterns of responses were similar. In general, soundscape questionnaires are administered *in situ* with pen and paper, while laboratory studies are more conducive to computer-based tasks. It is therefore fundamental to explore the transferability of results from one mode of administration to the other in the context of soundscape studies.

### Research questions

There are two bases for the present study on the ecological validity of soundscape reproduction. The first is that differences between soundscape ratings collected *in situ* and in laboratory settings is a relatively understudied domain considering its importance in the context of emerging audio technologies. It is important to establish if laboratory reproduction can elicit similar cognitive processes and reveal similar effects of contextual factors (such as time of day, day of week) on soundscape ratings. The second basis is the fact that the mode of administration (pen-and-paper vs. computer-based) has been found to influence ratings in other contexts, so one might wonder if it could also influence soundscape ratings.

As discussed above, a research setting can be considered *ecologically valid* only when three elements are present:

- the participants are representative of the studied population;

- the experimental setup and stimuli are representative of the studied environments;

- the experimental task and procedure are representative of the studied cognitive processes.

To answer the first requirement, little can be done outside of the recruitment procedure, in this case, by selecting participants familiar with the site of interest. The other two requirements are the focus of this paper addressing the research questions below.

At a theoretical level, comparing in situ and laboratory conditions:

1. Can similar effects of contextual factors (time of day, day of week, and location on site) on soundscape ratings be observed *in situ* and in the laboratory?

2. Can similar underlying soundscape dimensions be observed *in situ* and in the laboratory?

At a methodological level in laboratory settings:

3. Does the mode of administration influence soundscape ratings?

4. Does stimuli selection influence soundscape ratings?

### Methods

To answer the research questions, this study was structured in two connected parts. First, data was collected in a public space through a) users' questionnaire-based soundscape evaluations

and b) audio recordings taken during a representative portion of some of the data collection periods. Second, the audio recordings were reproduced in a laboratory experiment to collect participants' soundscape evaluations and compare those to the ones obtained on site. Ethical approval for this project was given by the Research Ethics Board II of McGill University [REB #686–0606 and #55–0615]. For the in situ study, participant consent was obtained verbally and participation details were reiterated and explained through a written (bilingual) description in the notebook containing the paper-based questionnaire–their written participation is taken as documentation of consent. For the laboratory experiments, participants signed a consent form and were compensated for their participation.

### *In situ* study

**Study site.** The study site was a small (about 1,800 m$^2$) public square in Montreal on one of the main commercial streets of that area (Avenue Mont-Royal), with shops and restaurants along two opposing traffic lanes also used by frequent bus lines (<10 minutes) during the day. On the far side from the commercial artery, the space is bordered by residences and a footpath. The locations of the recordings are shown in Fig 1.

**Participants.** *In situ*, participants were approached by a research team member while using the studied public space; generally, participants were only recruited if they had stopped in the space and spend at least a few minutes being exposed to the environment. They were asked to take a paper-based questionnaire, while the researcher noted the time of day and their location in the space while completing it.

Due to logistical constraints and evolving research considerations, the site study was not conducted with systematic factorial experimental design. Additionally, the location was not visited with the same frequency by users at all times of the day and week. Both of those factors

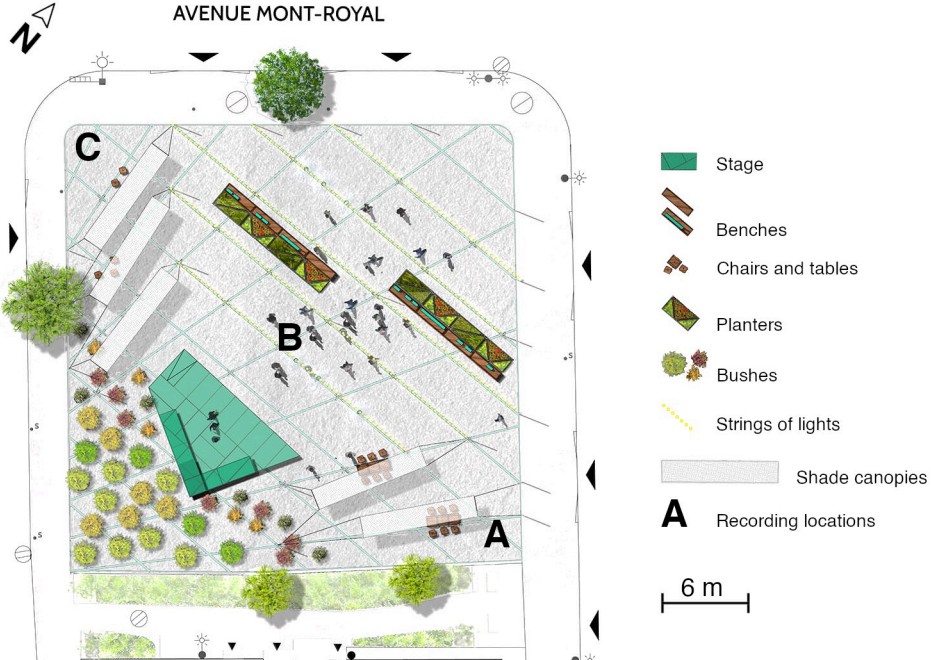

**Fig 1. Simplified map of the study site, showing recording locations A, B, and C.** Design layout provided by design firm Castor et Pollux and used and edited with permission.

**Table 1. Case counts [and sound levels in dBA ($LA_{eq,10min}$)] for each condition of in-situ data collection, separated by weekday-weekend, afternoon-evening, and noisy-quiet side of the space.**

| | Time period | | | | |
|---|---|---|---|---|---|
| | Weekday | | Weekend | | |
| Location | Afternoon | Evening | Afternoon | Evening | Total |
| Quiet | 15 [57.33] | 46 [61.45] | 5 [57.92] | 16 [58.69] | **82** |
| Noisy | 9 [62.69] | 53 [62.82] | 13 [62.46] | 28 [62.47] | **103** |
| **Total** | **24** | **99** | **18** | **44** | **185** |

led to highly unbalanced sample sizes in terms of weekday-weekend and afternoon-evening (Table 1). Despite variations in the visual design of the square, consistent with analyses conducted by Trudeau et al. [55], we collapse respondent data across the visual design conditions.

Afternoon and evening periods consisted of the time slots of 2 pm to 6 pm and after 6 pm, respectively. These choices, based on local working hours, and therefore activity levels, were confirmed by sound level trends on site [9]. Additionally, due to differential sound levels (see *Ambisonic recordings* section below), participants were grouped based on their location on site. The space was divided in half with a quiet and a noisy side, closest to the residential side and to the commercial street, respectively. Sample sizes for each condition are presented in Table 1.

A total of 185 questionnaires (102 women, 76 men, age = 34.76 ± 14.82) were collected. See a summary of average age, noise sensitivity (from the NSS scale [56]), and extraversion (from the BFI [57]–both collected with demographic questions at the end) in Table 2.

**Ambisonic recordings.** Ten-minute Ambisonic recordings were obtained with a Soundfield ST350 FOA (first-order Ambisonics) microphone and a Sound Devices 744T sound card at three locations on site (Fig 1). Sound levels were recorded simultaneously with a Brüel & Kjær type 2250 sound level meter. All recordings were obtained on the study site in September 2018.

Based on the 10-min average $LA_{eq}$ value for each recording, the two locations with the consistently lowest (range of 57.3–61.4 dBA) and highest (range of 61.9–66.5 dBA) sound levels were chosen for the laboratory experiments. The individual 10-min average $LA_{eq}$ values (see Table 1) were used to calibrate the reproduction levels in the listening room.

An additional recording session was conducted late at night on site to obtain a naturalistic background noise floor between conditions for the experiment, referred to as the *baseline* below.

## Laboratory study

**Participants.** For the laboratory studies, recruitment was conducted with the help of the Plateau borough in Montreal, to contact people who were familiar with the studied space,

**Table 2. Participants' profile for both laboratory studies (N = 20 and 14, respectively) and in situ (N = 185).**

| | Computer-based | | Pen-and-paper | | *In situ* | |
|---|---|---|---|---|---|---|
| | Mean | SD | Mean | SD | Mean | SD |
| **Age** | 44.60 | 16.61 | 45.93 | 17.09 | 34.76 | 14.82 |
| **Noise sensitivity** | 4.20 | 0.89 | 4.14 | 1.23 | 3.22 | 1.43 |
| **Extraversion** | 3.45 | 1.00 | 3.64 | 0.84 | 3.49 | 1.15 |

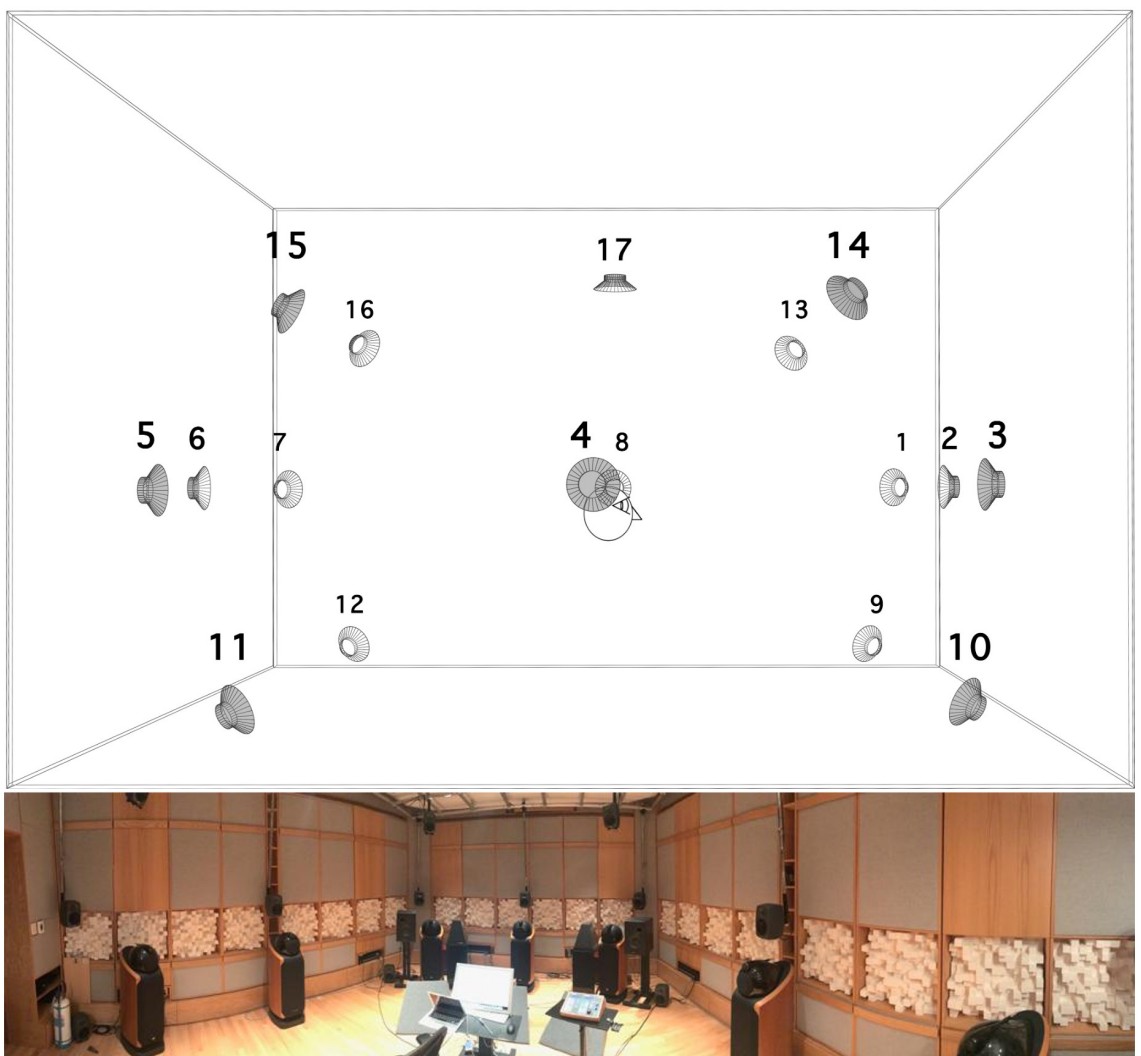

**Fig 2. Listening room.** Top: diagram of loudspeaker array from the side, simplified head for orientation; bottom: panoramic photograph of the room from the back right corner (original copyright: Grégoire Blanc [2019] under a CC BY 4.0 license).

whether living or working nearby. An official email from the borough was sent to their mailing list and a Facebook post was posted on their page on two occasions. A total of 34 people (adults with self-reported normal hearing) participated in both studies (Table 2): 20 for the computer-based study (8 women, age = 44.6 ± 16.6, 2 English), and 14 for the pen-and-paper study (10 women, age = 45.9 ± 17.1, 0 English). They received a compensation of 15$ for 1h30 of experiment.

**Conditions.** Two-minute excerpts were isolated from a subset of 10-minute long Ambisonic recordings chosen based on location in the space (locations A and C in Fig 1) and day and time of recording. Additionally, to investigate internal consistency, two excerpts were selected from each 10-minute recording. Conditions were selected in a factorial design, with 2 locations (quiet vs. noisy) × 2 days of the week (weekday vs. weekend) × 2 times of day (afternoon vs. evening) × 2 excerpts (selected 2-minute excerpts within each recording), for a total of 16 excerpts.

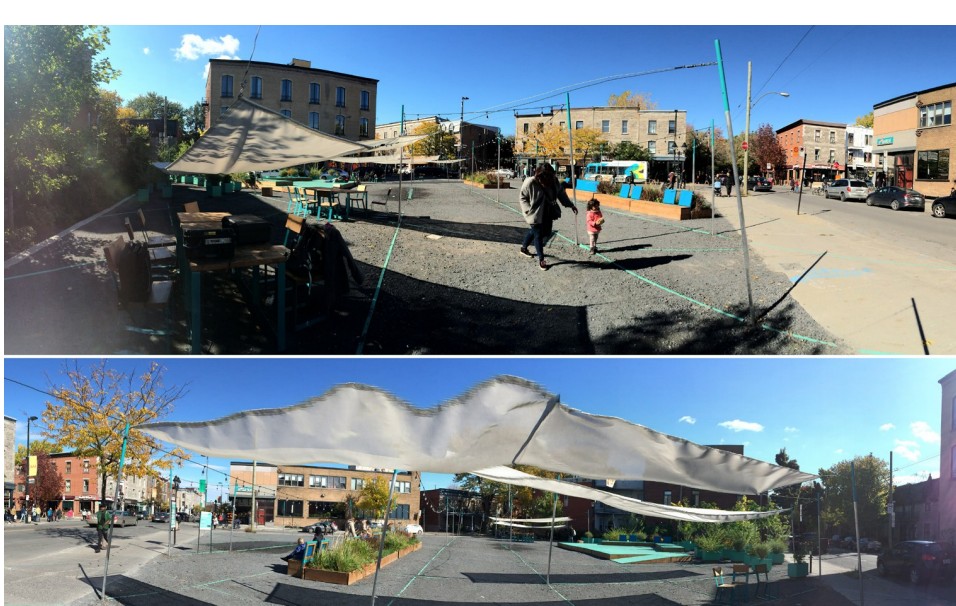

**Fig 3. Panoramic photographs of the space presented at the beginning of the laboratory experiments.** Top: location A; bottom: location C (original copyright: Mariana Mejía Ahrens [2018] under a CC BY 4.0 license).

**Procedure.** Participants were seated in the center of the listening room and loudspeaker array (Fig 2). Each trial lasted for 2 minutes, resulting in approximately 32 (2 minutes x 16 excerpts) minutes of testing with an optional break. They were first presented with two panoramic photographs of the studied site (the public space they were familiar with) from the two locations of recording facing the center of the space (Fig 3) for 30 seconds. They were then asked to listen to the 16 excerpts and fill out a shortened version of the questionnaire used *in situ* (Table 3). All excerpts were presented in a fully randomized order.

Within each trial, each excerpt was presented to the participants for 15 seconds before the questionnaire appeared to ensure they listened and acclimated to the soundscape. This was done to mirror the surveyed users of the studied site. They could answer the questionnaire for 1 min 40 s before the end of each excerpt. In the computer-based study, participants had no control over the timing of questionnaire presentation, all transitions were automated within the software. In the pen-and-paper study, participants were asked to respect this time, which appeared on screen, but the experimenter was not present to enforce it. However, all soundscape conditions (audio stimuli) were transitioned automatically for both studies. The last five seconds were used to fade into the baseline presented for 15 seconds between excerpts to avoid transitioning to silence.

The experimenter ran a practice trial with the participant before starting the experiment, to help them familiarize themselves with the task and automated timing. A short break was automatically triggered at the halfway point (after the 8th excerpt).

**Ambisonic reproduction.** Stimuli were presented in an acoustically-treated listening room (5.9 x 4.9 x 3.3 m) conforming to the ITU-R BS.775-1 standard [58] over an array of 17 Genelec 8030A loudspeakers placed on four height levels and facing the listener (Fig 2):

- a square of four at floor level (#9–12 in Fig 2)

- a square of eight located at head level (1.2 m above the floor–#1–8)

- a square of four suspended from the ceiling (2.3 m above the floor–#13–16)

- a single speaker directly above the listener (2.6 m above the floor–#17)

Decoding was conducted in MaxMSP version 8.0.5 (Cycling '74) using Heller's Ambisonic Decoder Toolbox for MATLAB [59] compiled for MaxMSP with Faust [60].

## Questionnaires

The full *in-situ* questionnaire used on site is the product of multiple iterations over the years and built on the literature presented in the *Soundscape assessment* section, using our Quebec French translation (see [61]). In this paper, we analyze the soundscape scales that were used in both in situ and laboratory conditions (Table 3). The question on *appropriateness* was rephrased to maintain equivalency between *in situ* and laboratory conditions, from: "I find this soundscape to be *appropriate for my activity*" (*in situ*) to: "I find this soundscape to be *appropriate for the activities I would conduct in this space*" (laboratory).

## Statistical analyses

To investigate the four research questions, we conducted three types of analyses:

1. to validate the dimensions underlying soundscape judgments [RQ2], we conducted a CFA on each data set (site and laboratory) with a model based on previous work [61],

2. to explore the influence of the mode of administration on soundscape assessments [RQ3], we followed up the CFA with an analysis of measurement invariance on the laboratory data,

3. to investigate the ecological validity of Ambisonic reproduction [RQ1] and the influence of stimuli choice [RQ3], we conducted a MANOVA on each data set (site and laboratory) with day, time, and location as independent variables for the site data, and day, time, location, mode of administration, and excerpt as independent variables for the laboratory data.

Further details are given for each analysis below.

Statistical analyses were computed in R 4.0.2 for Mac OS X and Rstudio® 1.3.1073, with $\alpha$ = 0.05. Both the laboratory and *in situ* data were highly non-normal, whether univariate or multivariate. The *in situ* data was additionally highly unbalanced with small groups (range of 5 to 53) when subdividing based on the three factors of interest: day (weekday-weekend), time

**Table 3. Questions for each of the 16 laboratory conditions.**

| Question | Type | Simplified name |
|---|---|---|
| I find this soundscape to be: | | |
| Pleasant | Likert scale | pleasant |
| Appropriate for the activities I would conduct in this space | Likert scale | appropriate |
| Monotonous | Likert scale | monotonous |
| Vibrant | Likert scale | vibrant |
| Chaotic | Likert scale | chaotic |
| Calm | Likert scale | calm |
| Eventful | Likert scale | eventful |
| Spending time in this soundscape gives me a break from my day-to-day routine: | Likert scale | restorative |

(afternoon-evening), and location (quiet-noisy). For these reasons, we chose to conduct semi-parametric analyses when pertinent.

Furthermore, missing values for Likert scales were replaced by the mean (rounded to 2 decimals) for each dependent variable per mode of administration in the laboratory (computer-based, pen-and-paper) and per visual design on site, as proportions of missing values were 6.5% or less (0.6–2.2% in the laboratory and 1.6–6.5% *in situ*). Because of this, we considered the Likert variables as continuous in the following analyses.

We first ran a Confirmatory Factor Analysis (CFA) on the site data to ensure that the latent dimensions did not differ from previous results obtained with the same questionnaire [61, 62]. We also conducted a CFA on the laboratory data with the same model, followed by an analysis of measurement invariance [63] between pen-and-paper and computer-based responses. Both CFA and measurement invariance were run on the laboratory data by accounting for repeated measures, as allowed by the *lavaan* package [64] for R. Measurement invariance testing consists of four steps: configural invariance, metric invariance, scalar invariance, and strict invariance (the latter is almost never needed and tested) [63]. Each step is more restrictive than, and relies on the validation of, the previous step. To validate a step, fit indices are compared to the ones from the previous, that is testing that the change in overall fit between two subsequent models falls under a certain threshold. A difference in Comparative Fit Index ($\Delta$CFI) $\leq$ 0.010 and a difference in Root Mean Square Error of Approximation ($\Delta$RMSEA) $\leq$ 0.015 are considered reasonably accurate to detect invariance for samples of more than 300 observations [65]. All CFA were conducted before replacement of missing data, using the *lavaan* package [64], and with the robust estimation method of Maximum Likelihood with Satorra-Bentler correction (MLM) due to the non-normality of the data [66].

A semi-parametric repeated-measure MANOVA with four within-subject factors (day, time, location, and excerpt) and one between-subject factor (mode of administration) was conducted on the laboratory data using the *multRM* function from the *MANOVA.RM* package, version 0.4.2 [67]. This package was developed to "enhance the small sample properties of [nonparametric procedures] while preserving their general applicability for all kinds of data in factorial repeated measures and split plot designs" [68]. Due to the covariance matrix being singular and the relatively small sample size, we used the Modified ANOVA-type statistic (MATS) and wild bootstrap resampling method for p-values, as recommended by the package authors [68]. The resampling was conducted with 1,000 iterations. Follow-up semi-parametric repeated-measure ANOVA with the same factors were conducted–with the *RM* function from the *MANOVA.RM* package looking at the ANOVA-type statistic (ATS) and wild bootstrap resampling–on each of the scales, with Šidák p-value corrections of $\alpha_{SID} = 0.0064$ for $\alpha = 0.05$.

Finally, a semi-parametric MANOVA, and follow-up semi-parametric ANOVA on each scale with Šidák p-value corrections of $\alpha_{SID} = 0.0064$ for $\alpha = 0.05$, with three factors (day, time, location) were conducted on the *in situ* data using the *MANOVA* function from the *MANOVA.RM* package. The ANOVA were not justified based on the MANOVA results but were conducted for comparison with the laboratory results.

## Results

The results are organized in three parts following the four research questions:

1. validating the dimensions underlying soundscape judgments [RQ2], with a CFA model based on previous work [61],

2. verifying methodological aspects of mode of administration with measurement invariance [RQ3] and of stimuli choice as a factor in the MANOVA [RQ4],

3. investigating the ecological validity of Ambisonic reproduction through the investigation of the effect of contextual factors in the MANOVA [RQ1].

## Dimensions underlying soundscape judgments

To investigate if the dimensions underlying participant's soundscape judgments, both *in situ* and in the laboratory, correspond to the previously found model [61, 62], we tested the same CFA model, which was as follows:

- "PL" factor measured by the variables "pleasant", "appropriate", "calm", "restorative", and "chaotic", representing the *pleasantness* dimension

- "EV" factor measured by the variables "eventful", "vibrant", "calm," and "chaotic", representing the *eventfulness* dimension.

***In situ* data.** The model fit for site data, was acceptable but not excellent, with $\chi^2_{SB}$ = 18.53, df = 11, p = 0.070; robust CFI = 0.970; robust RMSEA = 0.065, 90% CI [0.000, 0.121]; and SRMR = 0.054. In consequence, we looked at modification indices to explore how to improve the model, which suggested adding the correlation between "pleasant" and "appropriate" (mi = 11.035). This is supported by previous work [61, 62], wherein "pleasant" is consistently found to be associated with "appropriate". This new model yielded an excellent fit, with $\chi^2_{SB}$ = 10.19, df = 10, p = 0.424; robust CFI = 0.999; robust RMSEA = 0.011, 90% CI [0.000, 0.090]; and SRMR = 0.045, and was therefore retained.

The standardized estimates of the factor loadings in the improved model (Table 4) were middling to large (0.21–0.80) and statistically significant (all with p < 0.001 except "chaotic" on "EV" with p = 0.024). The "pleasant", "appropriate", "restorative", and "calm" variables loaded positively, while "chaotic" loaded negatively on the latent factor "PL". In parallel, "calm" loaded negatively, and "chaotic, "eventful", and "vibrant" loaded positively on the latent factor "EV".

Additionally, the two latent variables "PL" and "EV" are not strictly independent, showing a borderline significant (p = 0.069) but weak covariance (cov = 0.203, SE = 0.112), which is expected, as they share some measured variables. And finally, the added correlation between "pleasant" and "appropriate" is expectedly significant (p = 0.010) although moderate (cov = 0.363, SE = 0.079). Those results are very similar to those obtained in situ in previous studies [61, 62].

**Laboratory data.** In comparison to both the *in situ* data and previous results, we tested the same CFA model on the laboratory data. The model fit on the laboratory data was good,

**Table 4. Standardized factor loadings and standard errors (SE) for the retained CFA model for *in situ* data (N = 185).**

| Item | PL | | EV | |
|------|------|------|------|------|
| | Loadings | SE | Loadings | SE |
| **Pleasant** | 0.746 | 0.085 | | |
| **Appropriate** | 0.572 | 0.089 | | |
| **Restorative** | 0.604 | 0.095 | | |
| **Calm** | 0.795 | 0.096 | -0.365 | 0.106 |
| **Chaotic** | -0.563 | 0.101 | 0.210 | 0.111 |
| **Vibrant** | | | 0.637 | 0.128 |
| **Eventful** | | | 0.730 | 0.134 |

**Table 5. Factor loadings and standard errors (SE) for the retained CFA model for laboratory data (N = 544).**

| | PL | | EV | |
|---|---|---|---|---|
| Item | Loadings | SE | Loadings | SE |
| Pleasant | 0.904 | 0.069 | | |
| Appropriate | 0.869 | 0.087 | | |
| Restorative | 0.855 | 0.082 | | |
| Calm | 0.709 | 0.079 | -0.193 | 0.056 |
| Chaotic | -0.633 | 0.144 | 0.241 | 0.105 |
| Vibrant | | | 0.816 | 0.088 |
| Eventful | | | 0.794 | 0.104 |

with $\chi^2_{SB}$ = 23.87, df = 11, p < 0.05; robust CFI = 0.987; robust RMSEA = 0.068, 90% CI [0.030, 0.105]; and SRMR = 0.031, and was therefore retained, thus confirming that laboratory reproduction elicits similar latent dimensions to *in situ* listening.

The standardized estimates of the factor loadings in this model (Table 5) were middling to large (0.19–0.90) and statistically significant (p < 0.001). The "pleasant", "appropriate", "restorative", and "calm" variables loaded positively, while "chaotic" loaded negatively on the "PL" latent factor. In parallel, "calm" loaded negatively, and "chaotic", "eventful", and "vibrant" loaded positively on the "EV" latent factor.

Additionally, the two latent variables "PL" and "EV" are not strictly independent here as well, showing a significant (p = 0.001) but weak covariance (cov = -0.195, SE = 0.060).

**Comparison of latent dimensions between *in situ* and laboratory results.** Both CFA are validated, confirming that the previously developed model of factors underlying the soundscape ratings in our context is applicable for both *in situ* and laboratory data. The only difference between the two models is the addition of a correlation between pleasant and appropriate to the *in situ* model. This relation makes theoretical sense but we did not add it to the laboratory model in the interest of parsimony as it was already a good model. Comparing the two models' loadings, we see that the laboratory results are more salient, with higher absolute loading values and smaller standard errors.

## Methodological verifications

**Effect of mode of administration.** Following the validation of the CFA model on laboratory results, we tested the measurement invariance between modes of administration (pen-and-paper vs. computer-based). The first step, configural invariance, merely compares parameter estimates and p-values for the two groups of interest–pen-and-paper and computer-based. The model fit was acceptable to accept configural invariance (M1 in Table 6). The next step, metric invariance, forces identical factor loadings across groups. The change in model fit compared to the previous step was within bound, so we retained it (M2 in Table 6). The third step, scalar invariance, additionally forces identical intercepts between groups. The change in model fit compared to metric invariance was within bound and the model was retained (M3 in Table 6). The fourth, and last, step is strict invariance and constrains residuals in addition to the previous constraints. This change in model fit was too large and the model was not retained (M4 in Table 6), but this last step is rarely needed and tested.

Ultimately, our laboratory data showed scalar invariance between pen-and-paper and computer-based administration, with the exception of the intercept for "appropriate," which may necessitate more investigation to explain.

**Table 6. Tests of measurement invariance between pen-and-paper (N = 224) and computer-based (N = 320).**

| Model | $\chi^2_{SB}$ (df) | CFI | RMSEA (90% CI) | SRMR | comparison | $\Delta \chi^2_{SB}$ ($\Delta$ df) | p-value | $\Delta$ CFI | $\Delta$ RMSEA | $\Delta$ SRMR | Retain |
|---|---|---|---|---|---|---|---|---|---|---|---|
| **M1: configural** | 38.431 (22) | 0.985 | 0.072 (0.031–0.109) | 0.037 | – | – | – | – | – | – | Y |
| **M2: metric** | 45.456 (29) | 0.984 | 0.066 (0.023–0.101) | 0.059 | M2 vs. M1 | 8.1262 (7) | 0.322 | -0.006 | -0.001 | 0.022 | Y |
| **M3: scalar** | 56.557 (34) | 0.975 | 0.075 (0.038–0.108) | 0.065 | M3 vs. M2 | 9.9367 (5) | 0.077 | -0.013 | 0.011 | 0.006 | Y |
| **M4: strict** | 71.742 (41) | 0.963 | 0.083 (0.049–0.115) | 0.059 | M4 vs M3 | 13.665 (7) | 0.057 | -0.012 | -0.008 | 0.006 | N |

**Effect of stimuli choice.** The laboratory experiment relied on a factorial design that included the factor of excerpt selection. Within each recording of a specific combination of day, time, and location in the public space of interest, two distinct excerpts were selected to investigate the effect of excerpt selection. MANOVA results on laboratory data (see *Laboratory results* section below) showed no effect of excerpt. Excerpt was present in two significant interactions but they will not be detailed further since the main effect of excerpt was not significant and those interactions have no theoretical meaning.

## Ecological validity of laboratory reproduction

The following section details semi-parametric (M)ANOVA results using (modified) ANOVA-type statistics ((M)ATS)–between-subjects for *in situ* data and within-subjects for laboratory data–to compare the extent to which the same factors significantly moderate the data.

*In situ* **results.** *Overall MANOVA.* The semi-parametric independent MANOVA with day, time, and location as factors on the site data shows no main effects and no interactions (S1 Table).

*ANOVA per scale on site.* Unsurprisingly, following the MANOVA results, the follow-up ANOVA on each scale are all highly non-significant (S2 Table). These were conducted for the purpose of comparison with the laboratory results.

**Laboratory results.** *Overall MANOVA.* The repeated-measure MANOVA (Table 7) shows significant main effects of day (MATS = 14.93, p < 0.001), time (MATS = 42.13, p < 0.001), and location (MATS = 424.79, p < 0.001). Significant interactions between day

**Table 7. Modified ANOVA-type statistics (MATS) and their resampled p-values (wild bootstrap– 1,000 iterations) for RM MANOVA over all scales (N = 544).**

| | Test statistic | p-value |
|---|---|---|
| **Day** | **14.926** | **<0.001** |
| **Time** | **42.132** | **<0.001** |
| **Day x Time** | **7.381** | **0.026** |
| **Location** | **424.786** | **<0.001** |
| **Day x Location** | **24.967** | **<0.001** |
| **Time x Location** | **47.05** | **<0.001** |
| Day x Time x Location | 4.45 | 0.138 |
| Excerpt | 17.445 | 0.213 |
| **Day x Excerpt** | **30.555** | **<0.001** |
| Time x Excerpt | 2.415 | 0.723 |
| Day x Time x Excerpt | 2.735 | 0.487 |
| Location x Excerpt | 2.227 | 0.756 |
| **Day x Location x Excerpt** | **24.79** | **0.001** |
| Time x Location x Excerpt | 2.491 | 0.586 |
| Day x Time x Location x Excerpt | 2.474 | 0.58 |

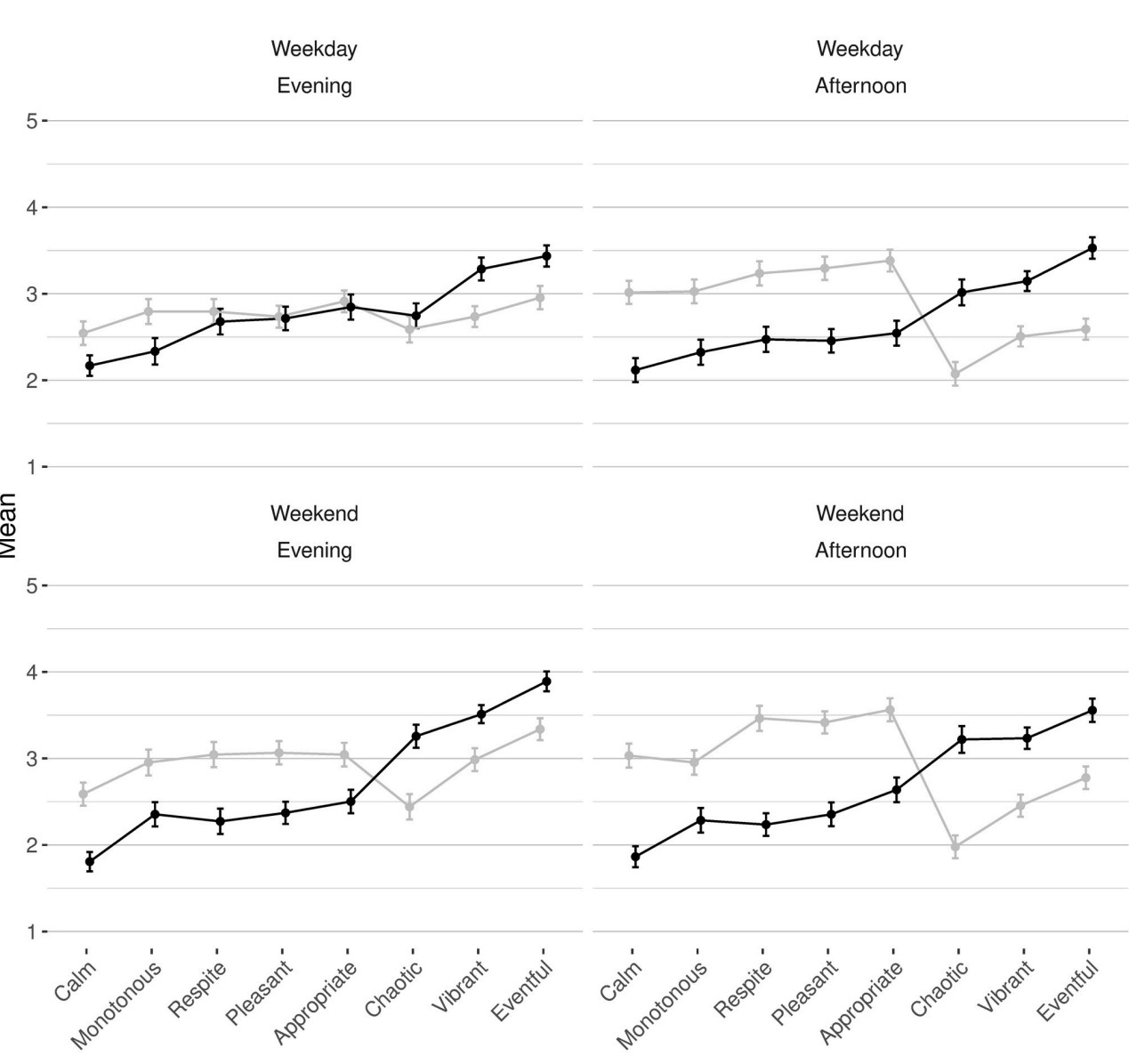

**Fig 4. Means and SE of all scales as a function of day, time, and location for the laboratory data (N = 544).**

and time (MATS = 7.38, p = 0.026), day and location (MATS = 24.97, p < 0.001), and time and location (MATS = 47.05, p < 0.001) were also found. Additional interactions, involving the excerpt, were found: day and excerpt (MATS = 30.55, p < 0.001), and day and location and excerpt (MATS = 24.79, p = 0.001). These will not be detailed further for the aforementioned reasons.

What is immediately evident from Fig 4 is that the same profile is found when comparing on the basis of location–comparing the corner of the public space closest to the residential area ("quiet" side) and the corner on the commercial street ("noisy" side). Pleasant, appropriate, monotonous, calm, and restorative are always higher in the quieter location, while chaotic,

vibrant, and eventful are always higher in the noisier location. The picture is less unequivocal for the effect of day of the week and time, so to understand those effects in a more granular manner, the next section describes post-hoc ANOVA with the same factors on each scale independently.

*ANOVA per scale in the lab*. The repeated-measure ANOVA show a significant main effect of location ($p < 0.001$) and no main effect of excerpt for all scales ($p > 0.0064$). Day has a main effect on eventful (ATS = 17.87, $p = 0.001$), and time has main effects on appropriate (ATS = 16.67, $p = 0.004$), vibrant (ATS = 21.47, $p < 0.001$), calm (ATS = 10.65, $p = 0.003$), eventful (ATS = 18.06, $p < 0.001$), and restorative (ATS = 21.47, $p < 0.001$).

The interaction of day and location is significant for pleasant (ATS = 15.06, $p < 0.001$) and chaotic (ATS = 14.12, $p = 0.001$), while the interaction of time and location is significant for pleasant (ATS = 12.60, $p = 0.002$), appropriate (ATS = 14.60, $p = 0.001$), chaotic (ATS = 16.97, $p < 0.001$), and calm (ATS = 14.99, $p < 0.001$). There are additional interactions involving the excerpt as well for those univariate ANOVA: day by excerpt for pleasant (ATS = 15.54, $p < 0.001$) and eventful (ATS = 16.21, $p < 0.001$).

Location is the most consistently significant factor with marked differences for all scales. Moving from the "quiet" side to the "noisy" side: pleasant, appropriate, monotonous, restorative and calm lose, while vibrant, eventful and chaotic gain, more than half a point (Fig 5). For the factor of time of day, appropriate, calm, and restorative decrease, while vibrant and eventful increase, by about a quarter of a point from afternoon to evening. Finally, day of the week has an effect only on eventful, with an increase between weekday and weekends of a quarter of a point as well.

The effect of location is further complicated by interactions, with the quiet location being found more pleasant during the weekend than during the week, but still always more so than the noisy side, despite the latter being found less pleasant during the weekend than the week. Meanwhile, the noisy side is evaluated as more chaotic during the weekend than weekdays, but always more chaotic than the quiet side, which sees no difference between weekend and weekdays (Table 8).

The quiet location also sees a difference between afternoons and evenings, being more pleasant, more appropriate, calmer, and less chaotic during afternoons, while the noisy side sees no differences (Table 9). One may have noticed that all those scales weigh in on the first CFA dimension of *pleasantness*, so a short summary could be to say that weekends and afternoons are more "pleasant" as a general umbrella concept than weekdays and evenings, while the effect of day is reversed and the effect of time is lost on the noisier side.

**Comparison of critical factors between *in situ* and laboratory results.** Contrary to our expectations, we did not see main effects of our factors of day, time, and location in the *in situ* data, not even from location, which is highly significant and markedly influential in the laboratory results. Those results hold both for the overall MANOVA and the post-hoc ANOVA on each scale. It is interesting to note that a visual comparison (Fig 5) reveals visible differences in ratings based on location, following similar patterns as the laboratory results: the quiet side is judged more pleasant and less eventful than the noisy side. However, the *in situ* differences between locations are not as wide as those in the laboratory study. These more pronounced results in the laboratory than *in situ* are reminiscent of the more salient factor loadings in the laboratory than *in situ* found in the *Dimensions underlying soundscape judgments* section. Fig 5 also shows that *in situ* results, regardless of location, are always more extreme than laboratory results for the scales contributing to the *pleasantness* dimension with higher ratings of pleasant, appropriate, calm and restorative, and lower ratings of chaotic.

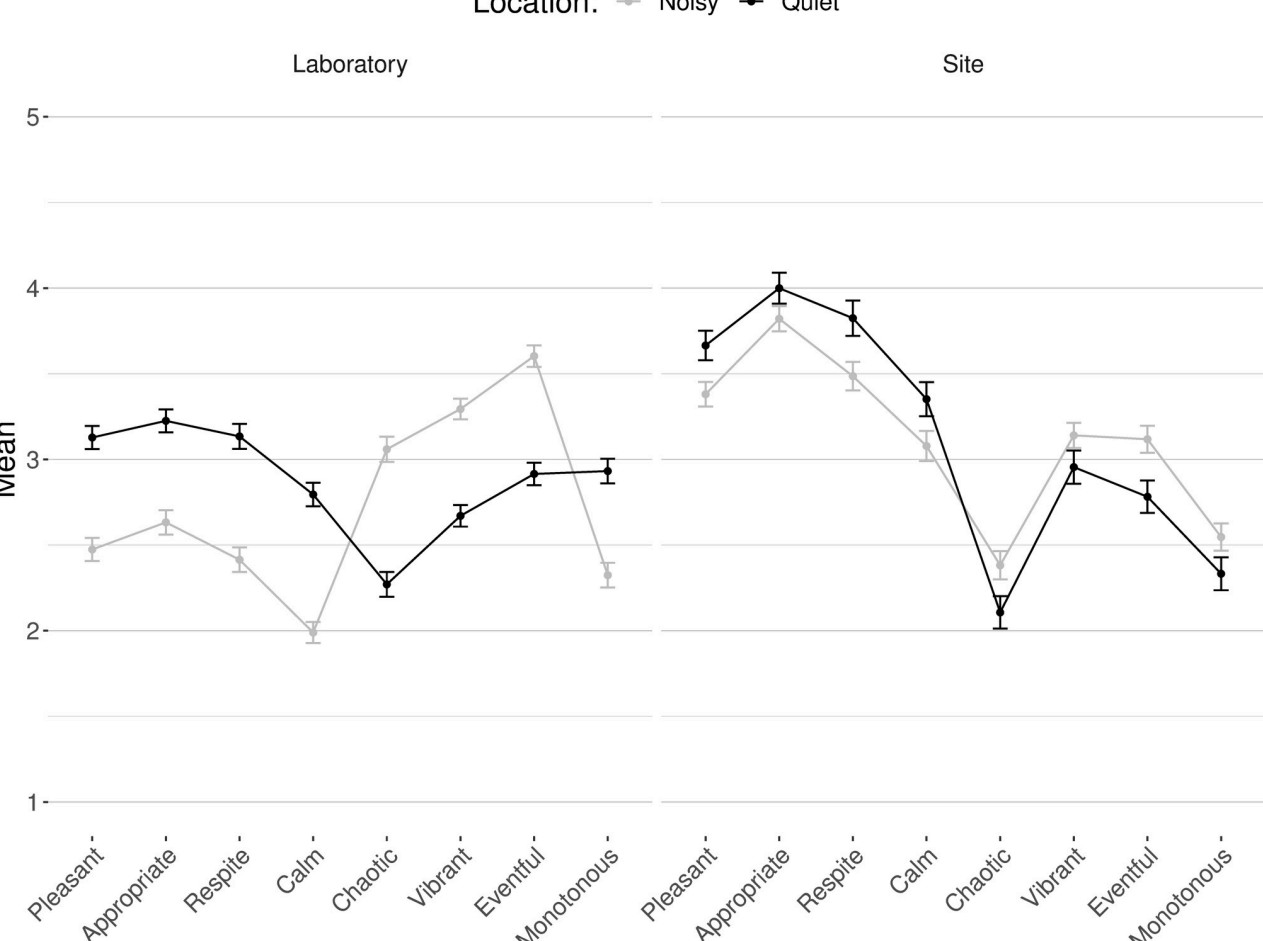

**Fig 5. Means and SE of all scales for each location in the laboratory (N = 544) and on site (N = 185).**

## Discussion

The driving force of this study is the objective of ensuring the ecological validity of soundscape reproduction and evaluation in the laboratory with the ultimate goal of bridging the gap between soundscape research and urban practice by increasing knowledge and developing tools to help urban professionals understand and imagine sound environments. Such a step opens the door for more quickly advancing and testing scientific theories, lowering the costs of mock-up designs, or using the laboratory as a communication space. However, before this step can be undertaken, the ecological validity of the methodological and technological choices

**Table 8. Means and SE for scales with significant interaction effect between location and day in the laboratory (N = 544).**

|  | Quiet | | Noisy | |
|---|---|---|---|---|
|  | **Weekday** | **Weekend** | **Weekday** | **Weekend** |
| **Pleasant** | 3.01 ± 1.11 | 3.24 ± 1.10 | 2.59 ± 1.12 | 2.36 ± 1.09 |
| **Chaotic** | 2.33 ± 1.21 | 2.21 ± 1.16 | 2.88 ± 1.21 | 3.24 ± 1.19 |

**Table 9. Means and SE for scales with significant interaction effect between location and time in the laboratory (N = 544).**

| | Quiet | | Noisy | |
|---|---|---|---|---|
| | **Afternoon** | **Evening** | **Afternoon** | **Evening** |
| **Pleasant** | 3.36 ± 1.08 | 2.90 ± 1.09 | 2.41 ± 1.12 | 2.54 ± 1.10 |
| **Appropriate** | 3.47 ± 1.07 | 2.98 ± 1.08 | 2.59 ± 1.18 | 2.67 ± 1.17 |
| **Calm** | 3.02 ± 1.12 | 2.57 ± 1.11 | 2.00 ± 1.08 | 1.99 ± 0.96 |
| **Chaotic** | 2.03 ± 1.10 | 2.51 ± 1.22 | 3.12 ± 1.25 | 3.00 ± 1.16 |

needs to be asserted by ensuring the representativity of the participants, of the setup and stimuli, and of the task and procedure.

In this study, we ensured that participants were representative of the population of interest by recruiting neighbors of the space. In the case of a future application for urban professionals, this may translate to different practice decisions which are already recommended and employed in general, from familiarizing themselves with the space from the perspective of the stakeholders using said space to co-creating with stakeholders.

The study therefore focused on the other two points by: 1) validating the dimensions underlying soundscape evaluations on site and in the laboratory [RQ2] 2) investigating the influence of the mode of administration of the questionnaire and of the specific portion of recording reproduced on judgments [RQ3-4] and 3) comparing judgments collected on-site and in the laboratory with the reproduced soundscape corresponding to the site of interest [RQ1].

## Validation of SSQP underlying dimensions

First, our findings confirm that participants hold similar dimensions of the underlying soundscape dimensions, as demonstrated by the CFA, both *in situ* and in the laboratory. The main difference was the additional correlation between pleasant and appropriate *in situ*, compared to the laboratory. The initial model was established in, and the additional correlation was supported by, previous work [61, 62]. Those results support the conclusion that 3D Ambisonic reproduction of soundscapes in the laboratory elicits similar latent dimensions as on-site listening in the use case of a small public space.

## Validation of methodological choices

Second, based on the validated CFA results, we found scalar measurement invariance between the two modes of administration tested in the laboratory (computer-based and pen-and-paper questionnaires). In other words, the way participants understand the soundscape items and use the measurement scale is similar between computer-based and pen-and-paper modes of administration. We intend to investigate this further by exploring open-ended responses about participants' understanding and use of the scales.

Additionally, the results of the analyses of variance on laboratory results showed no effect of the chosen excerpt within a 10-min recording. These results point to some level of freedom in procedure and stimuli choices.

## Influence of time, day, and location on soundscape evaluations

Finally, a direct statistical comparison between the data collected on site and in the laboratory was not possible by virtue of the experimental designs, so we examined (M)ANOVA results separately. On site, the analyses of variance showed no significant effects of the three contextual factors that were hypothesized to influence soundscape ratings [1, 6]: day of the week,

time of day, and location in the space, and their interactions. Previous work on the same site showed location, time of day, and day of the week influenced sound level, but did not look at soundscape evaluations [9].

In comparison, in the laboratory, the MANOVA showed a marked main effect of all three factors, as well as interactions of day and time, day and location, and time and location. Further explorations revealed that weekends were more eventful than weekdays, and afternoons were calmer and less eventful than evenings. Most markedly, location had a highly significant effect on all scales, which can be summarized as the quieter side being more pleasant and less eventful than the noisier side. This evident effect of location points to the need to record and reproduce multiple locations of any site of interest, even in this public space studied here, which was small, with traffic clearly audible at all locations in it.

Location also interacts with day and time, separately, in a way that can be summarized as the quiet side being more "pleasant" during the weekend, and during afternoons, while the noisy side is evaluated as less "pleasant" during the weekend, but not as a function of time of day. This latter interaction effect of time and location seems surprising given that the sound level on site was reported as higher during afternoons than evenings [9]. However location was not taken into account in [9], where long-term sound levels were obtained at only one point in the middle of the site.

Interestingly, a visual exploration (Fig 5) of the *in situ* results reveals consistently more extreme ratings of the variables of the first CFA dimension (*pleasantness*)–i.e., higher ratings of pleasant, appropriate, calm and restorative, and lower ratings of chaotic, regardless of location–on site in comparison to the laboratory results. This result might be due to a holistic integration of other sensory modalities and of contextual factors in the judgments on site. For example, expectations of space users towards the inevitability and rhythm of traffic noise in the soundscape [69] could have helped alleviate its effects in relation to time of day and day of the week–maybe even through deliberate choice of timing to use the space. This effect may be related to visual information, which has been shown to influence soundscape judgments, both in laboratory studies and *in situ* (see [70] for a review). Another potential element of response is the effect of laboratory stimuli, calibrated to match levels on site, being found to be louder than would be experienced on site [29]. Indeed, some participants in this study found the laboratory sound levels high, despite being told that levels were carefully matched to what they would, and did, experience on site as neighbors.

Furthermore, the same visualization (Fig 5) of the profile of responses in the laboratory depending on the location–although following the same profile as the *in situ* results, if less extreme–reveals much more pronounced differences between locations, as captured by the (M)ANOVA results. This points to the desired outcome of laboratory experimentation, wherein isolating the variables of interest makes it possible to reveal their effects. In this manner, our laboratory study allowed us to pull apart the different influences from sensory modalities and other contextual factors and to focus on the auditory modality and our factors of interest–namely day, time, location.

## Limitations and future directions

This study is a first step in confirming the ecological validity of 3D Ambisonic soundscape reproduction to collect soundscape evaluations in the laboratory. The results we obtained are in line with previous studies in soundscape research [36, 37, 71] pointing to the ecological validity of this technique for other tasks. In light of these encouraging results, we will extend our investigation to laboratory experiments manipulating other contextual factors, such as the activity at hand, on soundscape evaluations.

A limitation arising from the different experimental setting of the two studies is that participants in the laboratory were older on average. The different ages of participants between the laboratory study and the site study may have had an effect on soundscape evaluations [72], potentially in relation to higher noise sensitivity [61], via different hearing abilities (i.e., hearing loss).

Another experimental issue arose with the *in situ* data, wherein the factors investigated in this study emerged from the analysis of the *in situ* data, which were collected first. That is, during the initial *in situ* data collection, we did not systematically control for all the factors (not knowing which ones would be relevant) as we did in the lab using a factorial design (after the analysis of *in situ* data revealed relevant factors). As a result. sample sizes *in situ* were highly unbalanced with a range of 5 to 53 observations per condition (Table 1). As well, location on site was divided in two halves of the space, almost certainly aggregating observations from a gradient of sound experiences, while the recordings were captured at two opposite corners of the space. This could explain the more pronounced effects of location in laboratory settings.

Another point that may explain the lack of significant effects *in situ* is the holistic integration of other sensory modalities and of contextual factors in the judgments on site. In contrast, the laboratory experimental design did not present visual stimuli, as a deliberate choice for variable control, nor could it take into account other contextual factors such as the reason for visiting the space or the meaning attributed to the particular public square in the neighborhood. Finally, respondents on site were exposed to different soundscapes whereas, in the laboratory, all participants were presented with the exact same set of soundscapes and, as a result, on-site data could have more variation that we cannot account for. Similar concerns were raised by other work comparing *in situ* and laboratory soundscape judgments, though with binaural recordings [73]. We do intend to explore a way to account for such nuance by analyzing free-format questions about ambiance and sound sources audible in the soundscape collected both on site and in the laboratory experiment.

It is also interesting to note that the "monotonous" scale showed no main or interaction effects other than the effect of location, which may indicate a lack of clarity from the instrument as to the scale's meaning or applicability to soundscape judgments, in line with previous studies [61, 62, 74]. This is also a question we aim to look into specifically with additional data collected during this study with open-ended questions.

## Conclusions

To sum up, this study shows, on a theoretical level:

1. Marked effects of location, day of the week, and time of day were found in the laboratory, but not on site

2. 3D Ambisonic laboratory reproduction of soundscapes elicits similar latent dimensions than the equivalent *in situ* soundscapes

And on a methodological level:

3. mode of administration had little effect on soundscape evaluations in the laboratory

4. temporal variations within the same conditions (i.e., different excerpts from the same recording) seem to affect ratings little enough in comparison to the marked effects of location, day of the week, and time of the day.

An interesting finding that we did not foresee is that results on site seem to be much more "pleasant" in comparison to the laboratory results in general (i.e., including on the noisier side of the space), which hints at multiple possible cognitive processes, which potentially overlap.

This could be attributed to several reasons: that other sensory modalities integrate with the auditory perceptions to alleviate the unpleasantness of city noise, that the meaning of the space within the neighborhood (e.g., historical significance or break from urban landscape) may increase user satisfaction and with it soundscape pleasantness, and that people know and expect the city to be noisy and therefore employ conscious strategies to mitigate said noise– such as using the space at specific times. Another potential argument at play could be that the immersive reproduction of traffic noise is an uncomfortable reminder of how pervasive traffic is in the city by making it harder to ignore in a laboratory setting.

This study shows that laboratory soundscape studies confidently reproduce the patterns of *in situ* perceptions, and that this controlled setting allows one to magnify the effects of studied factors that can be lost in the variability of unconstrained *in situ* experience. This has implications for researchers, who need to be aware of this inflation of effects for its benefits and disadvantages both for research purposes as well as for the development of practical applications.

In particular, with regards to our goal of asserting the ecological validity of Ambisonic reproduction of soundscapes with the aim of developing a tool for urban professionals, awareness of the biases of this reproduction will be essential to sound urban practice. However, the present results plainly show highly similar soundscape latent dimensions between laboratory and on-site responses despite the clear difference in the amount of variability and nuance of respondent experience, justifying the adoption of Ambisonics for such urban practice tools.

Finally, this paper points to how important context is in two different ways: first, the straightforward results obtained in the laboratory study show the influences of the contextual factors of time, day, and location; second, the lack of effects on site reveals how much variability is introduced by the many cognitive processes at play in everyday life situations.

## Supporting information

**S1 Table. Modified ANOVA-type statistics (MATS) and their resampled p-values (wild bootstrap—1000 iterations) for MANOVA of on-site data (N = 185).**
(DOCX)

**S2 Table. Modified ANOVA-type statistics (MATS) for ANOVA (p-value resampled with wild bootstrap—1000 iterations) over each scale—No significant p-values (N = 185).**
(DOCX)

## Acknowledgments

The authors would like to thank the Arrondissement du Plateau-Mont-Royal for continued cooperation on this project, Mariana Mejía Ahrens for field recordings, Christopher Trudeau and Valérian Fraisse for assistance in data collection, and Grégoire Blanc for assistance in data collection and software design.

## Author Contributions

**Conceptualization:** Cynthia Tarlao, Daniel Steele, Catherine Guastavino.

**Data curation:** Cynthia Tarlao.

**Formal analysis:** Cynthia Tarlao.

**Funding acquisition:** Catherine Guastavino.

**Investigation:** Cynthia Tarlao, Daniel Steele.

**Methodology:** Cynthia Tarlao, Catherine Guastavino.

**Project administration:** Cynthia Tarlao, Catherine Guastavino.

**Resources:** Catherine Guastavino.

**Software:** Cynthia Tarlao.

**Supervision:** Catherine Guastavino.

**Visualization:** Cynthia Tarlao.

**Writing – original draft:** Cynthia Tarlao.

**Writing – review & editing:** Cynthia Tarlao, Daniel Steele, Catherine Guastavino.

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
