## [Decision Letter · Decision Letter 0]

13 Jan 2022

PONE-D-21-18471Assessing the ecological validity of soundscape reproduction in different laboratory settingsPLOS ONE

Dear Dr. Tarlao,

Thank you for submitting your manuscript to PLOS ONE. After careful consideration, we feel that it has merit but does not fully meet PLOS ONE’s publication criteria as it currently stands. Therefore, we invite you to submit a revised version of the manuscript that addresses the points raised during the review process. The manuscript has been evaluated by three reviewers, and their comments are available below. The reviewers have raised a number of major concerns, including  improvements to the reporting of methodological aspects of the study. Could you please carefully revise the manuscript to address all comments raised?Please submit your revised manuscript by Feb 26 2022 11:59PM. If you will need more time than this to complete your revisions, please reply to this message or contact the journal office at plosone@plos.org. Please include the following items when submitting your revised manuscript:A rebuttal letter that responds to each point raised by the academic editor and reviewer(s). You should upload this letter as a separate file labeled 'Response to Reviewers'.A marked-up copy of your manuscript that highlights changes made to the original version. You should upload this as a separate file labeled 'Revised Manuscript with Track Changes'.An unmarked version of your revised paper without tracked changes. You should upload this as a separate file labeled 'Manuscript'.If applicable, we recommend that you deposit your laboratory protocols in protocols.io to enhance the reproducibility of your results. Protocols.io assigns your protocol its own identifier (DOI) so that it can be cited independently in the future. For instructions see: https://journals.plos.org/plosone/s/submission-guidelines#loc-laboratory-protocols. Additionally, PLOS ONE offers an option for publishing peer-reviewed Lab Protocol articles, which describe protocols hosted on protocols.io. Read more information on sharing protocols at https://plos.org/protocols?utm_medium=editorial-email&utm_source=authorletters&utm_campaign=protocols.

We look forward to receiving your revised manuscript.

Kind regards,

Elisa Panada

Associate Editor

PLOS ONE

Journal Requirements:

2. We note that Figures 2 and 3 in your submission contain copyrighted images. All PLOS content is published under the Creative Commons Attribution License (CC BY 4.0), which means that the manuscript, images, and Supporting Information files will be freely available online, and any third party is permitted to access, download, copy, distribute, and use these materials in any way, even commercially, with proper attribution. For more information, see our copyright guidelines: http://journals.plos.org/plosone/s/licenses-and-copyright.

a. You may seek permission from the original copyright holder of Figures 2 and 3 to publish the content specifically under the CC BY 4.0 license. 

Reviewers' comments:

Reviewer's Responses to Questions

**Comments to the Author**

1. Is the manuscript technically sound, and do the data support the conclusions?

Reviewer #1: Yes

Reviewer #2: Partly

Reviewer #3: Yes

2. Has the statistical analysis been performed appropriately and rigorously? 

Reviewer #1: Yes

Reviewer #2: Yes

Reviewer #3: No

3. Have the authors made all data underlying the findings in their manuscript fully available?

Reviewer #1: Yes

Reviewer #2: Yes

Reviewer #3: Yes

4. Is the manuscript presented in an intelligible fashion and written in standard English?

Reviewer #1: Yes

Reviewer #2: No

Reviewer #3: Yes

5. Review Comments to the Author

Reviewer #1: Review

This is an interesting, important, and novel contribution to the field and it should be published. I outline some issues for consideration by the authors below.

Content

- “Context” is defined in L44 but this definition seems different from the one in L160 which refers to participants “state of mind” (whatever that means)

- L161 suggests that laboratory settings may _elicit_ different _perceptions_. But are “perceptions” elicited? So far as I understand from this claim, the authors refer to “responses” or “descriptions” or, maybe “experiences”.

- L195 The section is about “ecological approach to soundscape” (maybe better: “soundscapes”) but seems to be about “sounds” at least at first.

- L195 refers to Dubois asset action about how sounds are “perceived”. Perhaps worthwhile to doublecheck that work b/c so far as I recall, she is not focused on how sounds are perceived (i.e. physiologically experienced from a bottom-up perspective) but rather how they are experienced/understood/remembered/etc (i.e. from a top-down cognitive perspective) — this seems to be aligned with the summary described here.

- L197 Furthermore, please confirm the reference to “contextual information” (a problematic term in my opinion, and one you never use again) — what is “information” in this sense? Context is already a relatively vague concept, but adding “information” into the mix muddies the water here . Please verify in the original source.

- L203 Beware the concept of “attention” here. It’s unclear what kind of cognitive/physiological phenomena you refer to (“attention” is another word with a range of meanings in different disciplines). Maybe you mean “listen” (as opposed to “hear”).

- L242 For the first time the authors explain that they are interested in “conceptualizations” (the word shows up many times after this). This term comes out of nowhere for me, as a casual reader of this text. What is a conceptualization? What kind of phenomena is it and in what scientific domain is it studied? This feels ephemeral. Why not “descriptions”? Maybe this becomes clear later. Whatever you do just be consistent throughout. At other points you also talk about “eliciting cognitive processes”. Is this in addition to understanding “conceptualizations”? Even if this is my ignorance, the difference between “cognitive processes” and “conceptualizations” is vague. The term “process” is used often in this text and evidently in different ways (e.g. L187, is this in reference to cognitive processes or something else?) — if you agree, please verify all uses of this word are as intended

- L609 For the first time you raise the issue of helping urban professionals “understand and imagine”. Consider introducing this earlier than in the discussion. Evidently this is a key motivation.

- L716 The claim that “monotonous” results “indicate some confusion” may come across unintentionally condescending. I would flip this around: The issue is not with the participants’ so-called “confusion” but with the experiment design itself. One may wonder why “monotonous” was chosen in the first place — is it a term used by the cohort under investigation or a term chosen by scientists? Do participants get informed on what these terms mean (monotonous, soundscape, appropriate, …)? If not, then it seems unfair to label the participants as “confused”. S

- L733 “multiple possible processes, which potentially overlap” — I can’t understand this claim, is it the same use of “processes” in L756? Some more precision in both locations would be helpful

Style

- Section (sub-)headers could be more precise. This will help the reader better navigate the text. For instance, under “Introduction” is the the sub-header “Soundscape” (L32). I wonder if a header like “State-of-the-art in soundscape research” would be more useful. Similar issue is on L110 “Existing tools” (for what? Having a more descriptive title could help readers)

- L43 citations are in a different style

- L134 entails —> uses

- L134 and 2D —> and a 2D

- L182-184 this sentence is difficult to follow

- L196 “holisitc manner —> holistically

- L200 you may wish to enter a citation at the end of this line

- L205 Global manner —> holistically (check for all instances of “global” and correct)

- L212 3D, 1D, 2D, 2DFOA —> I don’t think these were previously defined. Do they warrant an explanation for readers outside of the field?

- L290 notated —> noted

- L361 “s” unit should follow a number on the same line

- L438-441 difficult to parse, consider rewriting

- L438 what is a “conceptualization”

- L473 than — > to

- L688 “as was presented” (just remove this)

- L689-690 split into two sentences to facilitate readability

- L708/L709 exposed vs presented — strange distinction here, I’m not sure if that’s intentional and consistent throughout this chapter

- Auditory scene / sound environment / soundscape >> please just use soundscape throughout since this is the term you introduce at the beginning. Otherwise it looks as if you are promoting some distinction.

Reviewer #2: Major comments:

The authors discussion on the ecological validity of laboratory data and in situ data through statistical methods is deeply impressive. But in terms of representation of participants, result is inevitably questionable. The average age difference between laboratory participants and in situ participants is too large (more than ten years old, almost a generation). It seems that the older participants showed less tolerance for background noise than younger adults. (Tun, P. A. ,1998). So the conclusion:” The young results on site seem to be much more

“pleasant” in comparison to the laboratory results.” might be influenced by the age difference.

Also, The authors did not report the hearing status of the participants, nor did they report the purpose of the on-site participants at this time (entertaining, commuting, or simply wandering around). At the same time, the gap in the number of participants should not be ignored (34 vs 185). It is necessary to conduct supplementary experiments with more and a wider range of participants. The demographic variables of participants needs to be reported in detail.

I suggest further revision on this paper.

Reference:

Tun, P. A. (1998). Fast noisy speech: age differences in processing rapid speech with background noise. Psychology and aging, 13(3), 424.

Minor comments:

Line 296 , there should be a reference to the work of Trudeau et al in 2020

Line 296: It is needed to clarify the meaning of “We collapse over all conditions.” Does it refer to the participants or other variables cannot be controlled in study site?

Table 1: Accurate decibel values between different scenes required.

Table 2: How did the author measure the noise sensitivity and extraversion of participants?

Line 395: Why there is missing values for Likert scales? The participants didn’t complete the questionnaire? Or are there some options that no one chose?

Table 3: “monotonous” Item doen’t show in the both CFA models. Why? Also “restorative” item only exists in the result part. The questionnaire doesn’t include it.

Reviewer #3: Ref. Review: PONE-D-21-18471

Paper Title: Assessing the ecological validity of soundscape reproduction in different laboratory Settings

Dear Authors and Editor,

Based on the text exposed in the entitled paper: "Assessing the ecological validity of soundscape reproduction in different laboratory Settings", I recommend major revisions before acceptance and publishing on the Plos One Journal.

The work presented in this paper proposed exploring the ecological validity of soundscape reproduction in the laboratory using first order Ambisonics and different modes of questionnaire administration.

The suggested work has the potential to be an excellent example of a listening test using first-order Ambisonics equipment for soundscape studies.

For the improvement of the manuscript quality, the following suggestions should be observed:

Methods:

Procedure:

1. There was no training period. How to be sure if the participants understood all asked tasks in each experiment procedure?

2. Regarding the time control in the computer-based experiment, the authors informed that the time was not controlled on the same basis in the paper-and-pen experiment. For sure, the time control adds some stress to the listeners and could be a bias in this experiment. The optimal situation should be adding time control for the paper-and-pen experiment or removing the control of time for the computer-based experiment. Could you please check the mentioned bias and ensure that the related research question will show reliable results?

Statistical analysis:

3. Could the author show a flowchart with all statistical procedures, indicating how the statistical analysis was used for each research question and expected output?

Results:

4. What about the CFA assumptions? Could you please inform in the text if all assumptions to conduct a CFA were followed?

https://www.statisticssolutions.com/free-resources/directory-of-statistical-analyses/confirmatory-factor-analysis/

5. Could you please also show if the MANOVA assumptions were fulfilled?

6. PLOS authors have the option to publish the peer review history of their article (what does this mean?). If published, this will include your full peer review and any attached files.

Reviewer #1: No

Reviewer #2: **Yes: **Yanqun Yang

Reviewer #3: No

---

## [Author Response · Author response to Decision Letter 0]

11 Apr 2022

The authors thank the reviewers for their comments, which have greatly improved the quality of the manuscript. We took them into consideration and trust that we adequately addressed the concerns raised in this rebuttal letter (red font) and by the changes made to the manuscript (yellow highlights). We proceed with dealing with all the individual comments in a point-by-point fashion.

We also took into account the editor’s comment about journal requirements. Specifically regarding photo copyright, we added information in the captions of Figures 2 and 3 indicating that the photos credited were taken by members of the team who agree to the CC BY 4.0 license. 

Reviewer #1: 

Review

This is an interesting, important, and novel contribution to the field and it should be published. I outline some issues for consideration by the authors below.

Content

- “Context” is defined in L44 but this definition seems different from the one in L160 which refers to participants “state of mind” (whatever that means)

 We have rephrased “state of mind” for clarity (L171-172). This sentence now reads “Laboratory settings differ from everyday life situations and therefore may elicit different judgments, whether through different perceptions, experiences, expectations, or biases, specifically in terms of contextual factors (for example, the reason for choosing to visit a particular space at a particular time).”

- L161 suggests that laboratory settings may _elicit_ different _perceptions_. But are “perceptions” elicited? So far as I understand from this claim, the authors refer to “responses” or “descriptions” or, maybe “experiences”.

 We would argue that yes, perceptions are elicited. However, we have rephrased this sentence to clarify and give a more inclusive, yet specific description of what we meant (L170-172) as follows “Laboratory settings differ from everyday life situations and therefore may elicit different judgments, whether through different perceptions, experiences, expectations, or biases, specifically in terms of contextual factors (for example, the reason for choosing to visit a particular space at a particular time).”.

- L195 The section is about “ecological approach to soundscape” (maybe better: “soundscapes”) but seems to be about “sounds” at least at first.

 The notions of environmental sounds and everyday listening are precursors of the development of the soundscape research field and are mentioned for historical/chronological accuracy but they do not constitute the core or the majority of this section. We therefore feel that the heading is correctly formulated.

- L195 refers to Dubois asset action about how sounds are “perceived”. Perhaps worthwhile to doublecheck that work b/c so far as I recall, she is not focused on how sounds are perceived (i.e. physiologically experienced from a bottom-up perspective) but rather how they are experienced/understood/remembered/etc (i.e. from a top-down cognitive perspective) — this seems to be aligned with the summary described here.

 Dubois talks about the two-way interaction between conceptualization (top-down) and perception (bottom-up). We therefore feel that our phrasing (now in L208-209) is accurate.

- L197 Furthermore, please confirm the reference to “contextual information” (a problematic term in my opinion, and one you never use again) — what is “information” in this sense? Context is already a relatively vague concept, but adding “information” into the mix muddies the water here . Please verify in the original source.

 We have removed the confusing term ‘contextual information’ and rephrased for clarity and specificity (L209-210). The sentence now reads “Indeed, Dubois [48] showed that sounds are also perceived and identified holistically by listeners who integrate everyday situations in which the sounds are experienced into complex mental representations [49].”.

- L203 Beware the concept of “attention” here. It’s unclear what kind of cognitive/physiological phenomena you refer to (“attention” is another word with a range of meanings in different disciplines). Maybe you mean “listen” (as opposed to “hear”).

 Thank you, we have rephrased to avoid confusion (now L216).

- L242 For the first time the authors explain that they are interested in “conceptualizations” (the word shows up many times after this). This term comes out of nowhere for me, as a casual reader of this text. What is a conceptualization? What kind of phenomena is it and in what scientific domain is it studied? This feels ephemeral. Why not “descriptions”? Maybe this becomes clear later. Whatever you do just be consistent throughout. At other points you also talk about “eliciting cognitive processes”. Is this in addition to understanding “conceptualizations”? Even if this is my ignorance, the difference between “cognitive processes” and “conceptualizations” is vague. The term “process” is used often in this text and evidently in different ways (e.g. L187, is this in reference to cognitive processes or something else?) — if you agree, please verify all uses of this word are as intended

 Thank you, we have changed conceptualizations and clarified “cognitive” processes when accurate throughout the text.

- L609 For the first time you raise the issue of helping urban professionals “understand and imagine”. Consider introducing this earlier than in the discussion. Evidently this is a key motivation.

 Thank you for pointing this out, we have emphasized this point in the introduction (see L88-92)

- L716 The claim that “monotonous” results “indicate some confusion” may come across unintentionally condescending. I would flip this around: The issue is not with the participants’ so-called “confusion” but with the experiment design itself. One may wonder why “monotonous” was chosen in the first place — is it a term used by the cohort under investigation or a term chosen by scientists? Do participants get informed on what these terms mean (monotonous, soundscape, appropriate, …)? If not, then it seems unfair to label the participants as “confused”. 

 Thank you, this is a very valid point. The soundscape items we use were developed based on participants spontaneous descriptions (Axelsson et al., (2010)) and have been codified into the ISO definition of soundscape methodology. The term is generally not explained/defined in the process of assessment as deployed by soundscape researchers. We have reframed this point, as we agree the onus is on the researcher to pose clear questions and tasks to our participants (see L750). The sentence now reads “[…] which may indicate a lack of clarity from the instrument as to the scale’s meaning or applicability to soundscape judgments […]”.

- L733 “multiple possible processes, which potentially overlap” — I can’t understand this claim, is it the same use of “processes” in L756? Some more precision in both locations would be helpful

 We have specified all types of processes mentioned throughout the manuscript, including those two instances which refer to cognitive processes (now on L767 and L790 for those two instances).

Style

- Section (sub-)headers could be more precise. This will help the reader better navigate the text. For instance, under “Introduction” is the the sub-header “Soundscape” (L32). I wonder if a header like “State-of-the-art in soundscape research” would be more useful. Similar issue is on L110 “Existing tools” (for what? Having a more descriptive title could help readers)

 We have clarified the two sub-headings mentioned above, as well as the sub-heading about soundscape assessment.

- L43 citations are in a different style

corrected

- L134 entails —> uses

corrected (L144)

- L134 and 2D —> and a 2D

corrected to “2D panoramic pictures” (L44)

- L182-184 this sentence is difficult to follow

 rephrased for clarity (L195)

- L196 “holisitc manner —> holistically

corrected (L209)

- L200 you may wish to enter a citation at the end of this line

done (now L213)

- L205 Global manner —> holistically (check for all instances of “global” and correct)

corrected (L218)

- L212 3D, 1D, 2D, 2DFOA —> I don’t think these were previously defined. Do they warrant an explanation for readers outside of the field?

We have added a footnote to clarify 1D, 2D, 3D (referenced on L225) and we have removed superfluous mentions to FOA in the paper for clarity (i.e., except in the methods section for an accurate description of the audio methodology).

- L290 notated —> noted

corrected (L302)

- L361 “s” unit should follow a number on the same line

corrected (L375)

- L438-441 difficult to parse, consider rewriting

 simplified and rephrased (L466-468). The sentence now reads “To investigate if the dimensions underlying participant’s soundscape judgments, both in situ and in the laboratory, correspond to the previously found model [61,62], we tested the same CFA model, which was as follows”.

- L438 what is a “conceptualization”

 we have changed that word throughout to “(latent) dimensions”

- L473 than — > to

corrected (L500)

- L688 “as was presented” (just remove this)

corrected (L717)

- L689-690 split into two sentences to facilitate readability 

 The sentence was mistakenly divided by commas, which we have corrected (L718-720)

- L708/L709 exposed vs presented — strange distinction here, I’m not sure if that’s intentional and consistent throughout this chapter

We make the distinction because of the “natural” experience on site vs the “artificial” presentation of recordings in the lab (now L742-744)

- Auditory scene / sound environment / soundscape >> please just use soundscape throughout since this is the term you introduce at the beginning. Otherwise it looks as if you are promoting some distinction.

In general, we do make a distinction; however, as it’s not central to the point of the study, we have removed any new terms that may surprise the reader throughout the manuscript.

 

Reviewer #2: 

Major comments:

The authors discussion on the ecological validity of laboratory data and in situ data through statistical methods is deeply impressive. But in terms of representation of participants, result is inevitably questionable. The average age difference between laboratory participants and in situ participants is too large (more than ten years old, almost a generation). It seems that the older participants showed less tolerance for background noise than younger adults. (Tun, P. A. ,1998). So the conclusion:” The young results on site seem to be much more

“pleasant” in comparison to the laboratory results.” might be influenced by the age difference.

Also, The authors did not report the hearing status of the participants, nor did they report the purpose of the on-site participants at this time (entertaining, commuting, or simply wandering around).

Thank you for your comments on our participants. To the extent possible, we make amendments to the manuscript, otherwise, we comment directly here. First, we added a paragraph about age and hearing status (L723-727) in the discussion. Our recruitment email for the laboratory study specified we sought people with self-reported normal hearing (which we now indicate in the manuscript – L342-343) but, based on our existing ethics review, we could not and did not seek permission to ask about health information, especially while doing a questionnaire with people in the public space.

Concerning the purpose of on-site participants for visiting the space, this information was collected, but like another number of questions in the on-site questionnaire, it was not analyzed in this paper since it was not reproduced in the laboratory and is therefore not comparable. We do mention exploring this element in the future (under the notion of “activity” – see L721).

At the same time, the gap in the number of participants should not be ignored (34 vs 185). It is necessary to conduct supplementary experiments with more and a wider range of participants.

 The 34 participants all listened to the 16 laboratory conditions (for a total of 544 data points) in a repeated-measure design, while the 185 participants on site were independent measures. Regardless, the statistical analyses are conducted independently between studies.

The demographic variables of participants needs to be reported in detail.

 Demographic variables are not investigated in this paper and thus not presented further.

I suggest further revision on this paper.

Reference:

Tun, P. A. (1998). Fast noisy speech: age differences in processing rapid speech with background noise. Psychology and aging, 13(3), 424.

 In this study, we do not present speech in noise and do not measure any type of performance, but we now cite pertinent work on the topic from the field of soundscape (refs. [61] and [72] cited on L725-726).

Minor comments:

Line 296 , there should be a reference to the work of Trudeau et al in 2020

corrected (L309)

Line 296: It is needed to clarify the meaning of “We collapse over all conditions.” Does it refer to the participants or other variables cannot be controlled in study site?

Thank you, we clarified this sentence, which now reads “[…] we collapse respondent data across the visual design conditions.” (L309)

Table 1: Accurate decibel values between different scenes required.

We have added the exact LAeq,10min values for each recording in table 1 and referenced it on L334.

Table 2: How did the author measure the noise sensitivity and extraversion of participants?

These assessments are well-established in the literature (NSS (Benfield et al., 2012) and BFI (Gosling et al., 2003)) and are not further analyzed in this study, so we did not go into more details for the sake of brevity. We added the references in the text (L321-322).

Line 395: Why there is missing values for Likert scales? The participants didn’t complete the questionnaire? Or are there some options that no one chose?

Missing values were calculated and replaced by scale (“for each dependant variable” – now L422), hence a range of values per study. If participants had not completed the questionnaire, they would not have been counted as participants, but some participants skipped some of the scales (whether by accident or due to not knowing/wanting to answer them).

Table 3: “monotonous” Item doen’t show in the both CFA models. Why? Also “restorative” item only exists in the result part. The questionnaire doesn’t include it.

“monotonous” could not be taken into account in the CFA model due to identification constraints (as explained in Tarlao et al., 2021, i.e., ref. [61])

“restorative” is the question about “break from daily routine” � we have edited table 3 to reflect that

 

Reviewer #3: 

Ref. Review: PONE-D-21-18471

Paper Title: Assessing the ecological validity of soundscape reproduction in different laboratory Settings

Dear Authors and Editor,

Based on the text exposed in the entitled paper: "Assessing the ecological validity of soundscape reproduction in different laboratory Settings", I recommend major revisions before acceptance and publishing on the Plos One Journal.

The work presented in this paper proposed exploring the ecological validity of soundscape reproduction in the laboratory using first order Ambisonics and different modes of questionnaire administration.

The suggested work has the potential to be an excellent example of a listening test using first-order Ambisonics equipment for soundscape studies.

For the improvement of the manuscript quality, the following suggestions should be observed:

Methods:

Procedure:

1. There was no training period. How to be sure if the participants understood all asked tasks in each experiment procedure?

 During the laboratory study, all participants were first presented a practice trial in the presence of the experimenter to ensure they understood the task and how to answer correctly (see L382-383), but no further training was implemented so as to more closely parallel the on-site procedure. The on-site procedure is based on established soundscape assessment scales which do not require training, as they aim to capture perceptions rather than performance. Further, with our piloting procedure, we found that the time given to participants was adequate for them to think and reflect on each of the items they were reporting on.

2. Regarding the time control in the computer-based experiment, the authors informed that the time was not controlled on the same basis in the paper-and-pen experiment. For sure, the time control adds some stress to the listeners and could be a bias in this experiment. The optimal situation should be adding time control for the paper-and-pen experiment or removing the control of time for the computer-based experiment. Could you please check the mentioned bias and ensure that the related research question will show reliable results?

 In the pen-and-paper experiment, the timing was controlled by the automatic transitioning between conditions by the software – this was the same as for the computer-based participants (we have edited the manuscript to make this clearer – see L377 and L379-380). We clarify that the key difference is that we could not take the paper questionnaires away from the participants as soon as each trial ended. All participants had correctly filled out the full number of sheets corresponding to each condition and none reported any difficulty in following the procedure. Further, the timings for each trial were piloted beforehand with 5 participants who were familiar with neither the software nor the task, to ensure participants had plenty of time to answer. No participants indicated that they were stressed for time in responding to any of the conditions.

Statistical analysis:

3. Could the author show a flowchart with all statistical procedures, indicating how the statistical analysis was used for each research question and expected output?

 We have clarified the sequence of analyses in the beginning of the statistical analyses section (L405-413).

Results:

4. What about the CFA assumptions? Could you please inform in the text if all assumptions to conduct a CFA were followed?

 We indicate in the methods that our data did not meet multivariate normality (L416), and that N(site) = 185 (L320), and N(lab) = 34 participants (L342) x16 conditions (L352). Additionally, the CFA model and the process to obtain it was described in a previous paper referenced in the relevant paragraphs (L407, L427 and L467 – ref. [61]). 

 However, we are deeply thankful for your comment, as it led us to re-check our CFA and measurement invariance analysis, upon which we realized an oversight regarding repeated measures which has now been corrected (L429-431, L497-498, L528-532, and table 6). The new, corrected analysis actually shows a better fit than the previous draft.

https://www.statisticssolutions.com/free-resources/directory-of-statistical-analyses/confirmatory-factor-analysis/

5. Could you please also show if the MANOVA assumptions were fulfilled?

 The semi-parametric repeated-measures MANOVA we used does not rely on assumptions (see Friedrich et al., 2017, i.e., ref. [68]).

---

## [Decision Letter · Decision Letter 1]

16 May 2022

PONE-D-21-18471R1Assessing the ecological validity of soundscape reproduction in different laboratory settingsPLOS ONE

Dear Dr. Tarlao,

Thank you for submitting your manuscript to PLOS ONE. After careful consideration, we feel that it has merit but does not fully meet PLOS ONE’s publication criteria as it currently stands. Therefore, we invite you to submit a revised version of the manuscript that addresses the points raised during the review process.

The reviewers largely feel that you have addressed their comments in your latest revision; however, Reviewer 3 has raised a few minor points that must be addressed. These include a request for clarification regarding statistics, specifically the MANOVA results. The reviewers full comments can be found below and in the attached file.

We look forward to receiving your revised manuscript.

Kind regards,

Natasha McDonald, PhD

Associate Editor

PLOS ONE

Journal Requirements:

Reviewers' comments:

Reviewer's Responses to Questions

**Comments to the Author**

1. If the authors have adequately addressed your comments raised in a previous round of review and you feel that this manuscript is now acceptable for publication, you may indicate that here to bypass the “Comments to the Author” section, enter your conflict of interest statement in the “Confidential to Editor” section, and submit your "Accept" recommendation.

Reviewer #1: All comments have been addressed

Reviewer #2: All comments have been addressed

Reviewer #3: All comments have been addressed

2. Is the manuscript technically sound, and do the data support the conclusions?

Reviewer #1: Yes

Reviewer #2: Yes

Reviewer #3: Yes

3. Has the statistical analysis been performed appropriately and rigorously? 

Reviewer #1: Yes

Reviewer #2: Yes

Reviewer #3: Yes

4. Have the authors made all data underlying the findings in their manuscript fully available?

Reviewer #1: Yes

Reviewer #2: Yes

Reviewer #3: Yes

5. Is the manuscript presented in an intelligible fashion and written in standard English?

Reviewer #1: Yes

Reviewer #2: Yes

Reviewer #3: (No Response)

6. Review Comments to the Author

Reviewer #1: Excellent paper, important results, rigorous science. A pleasure to read this. Congratulations. Looking forward to seeing where you go next with this.

Reviewer #2: The authors have answered all my questions and I think this manuscript suitable for publication now.

Reviewer #3: Ref. Review: PONE-D-21-18471R1

Paper Title: Assessing the ecological validity of soundscape reproduction in different laboratory Settings

Dear Authors and Editor,

Based on the text exposed in the entitled paper: "Assessing the ecological validity of soundscape reproduction in different laboratory Settings", I recommend minor revisions before acceptance and publishing in the Plos One Journal.

The work presented in this paper proposed exploring the ecological validity of soundscape reproduction in the laboratory using first-order Ambisonics and different modes of questionnaire administration.

The suggested work has the potential to be an excellent example of a listening test using first-order Ambisonics equipment for soundscape studies.

For the improvement of the manuscript quality, the following suggestions should be observed:

Results:

Could you please also show if the MANOVA assumptions were fulfilled?

Authors' answer: The semi-parametric repeated-measures MANOVA we used does not rely on assumptions (see Friedrich et al., 2017, i.e., ref. [68]).

Could you please complement the text with the argumentation of the ref [68]? For example, I suggest:

"preserving their general applicability for all kinds of data in factorial repeated measures and split-plot designs" [68], eliminating the MANOVA assumptions fulfilment.

7. PLOS authors have the option to publish the peer review history of their article (what does this mean?). If published, this will include your full peer review and any attached files.

Reviewer #1: **Yes: **Matt Coler

Reviewer #2: **Yes: **Yanqun Yang

Reviewer #3: **Yes: **Margret Sibylle Engel

---

## [Author Response · Author response to Decision Letter 1]

18 May 2022

The authors thank the reviewers for their comments. We trust that we adequately addressed the last remaining concern mentioned (see tracked changes in the manuscript).

---

## [Editor Report · Decision Letter 2]

10 Jun 2022

Assessing the ecological validity of soundscape reproduction in different laboratory settings

PONE-D-21-18471R2

Dear Dr. Tarlao,

We’re pleased to inform you that your manuscript has been judged scientifically suitable for publication and will be formally accepted for publication once it meets all outstanding technical requirements.

Kind regards,

Carla Pegoraro

Division Editor

PLOS ONE

---

## [Editor Report · Acceptance letter]

16 Jun 2022

PONE-D-21-18471R2 

Assessing the ecological validity of soundscape reproduction in different laboratory settings 

Dear Dr. Tarlao:

I'm pleased to inform you that your manuscript has been deemed suitable for publication in PLOS ONE. Congratulations! Your manuscript is now with our production department. 

Kind regards, 

on behalf of

Dr Carla Pegoraro 

Staff Editor

PLOS ONE